# Breaking Entry-and Species Barriers: LentiBOOST^®^ Plus Polybrene Enhances Transduction Efficacy of Dendritic Cells and Monocytes by Adenovirus 5

**DOI:** 10.3390/v14010092

**Published:** 2022-01-05

**Authors:** Astrid Strack, Andrea Deinzer, Christian Thirion, Silke Schrödel, Jan Dörrie, Tatjana Sauerer, Alexander Steinkasserer, Ilka Knippertz

**Affiliations:** 1Department of Immune Modulation, Universitätsklinikum Erlangen, Friedrich-Alexander Universität Erlangen-Nürnberg, Hartmannstr. 14, 91052 Erlangen, Germany; andrea.deinzer@uk-erlangen.de (A.D.); alexander.steinkasserer@uk-erlangen.de (A.S.); 2Institute of Clinical Microbiology, Immunology and Hygiene, Universitätsklinikum Erlangen, Friedrich-Alexander Universität Erlangen-Nürnberg, Wasserturmstraße 3/5, 91054 Erlangen, Germany; 3SIRION Biotech GmbH, Am Klopferspitz 19, 82152 Martinsried, Germany; Thirion@sirion-biotech.de (C.T.); Schroedel@sirion-biotech.de (S.S.); 4Department of Dermatology, Universitätsklinikum Erlangen, Friedrich-Alexander Universität Erlangen-Nürnberg, Hartmannstr. 14, 91052 Erlangen, Germany; Jan.Doerrie@uk-erlangen.de (J.D.); tatjana.sauerer@uk-erlangen.de (T.S.)

**Keywords:** adenovirus, adenoviral transduction, monocytes, dendritic cells, T cells, LentiBOOST^®^

## Abstract

Due to their ability to trigger strong immune responses, adenoviruses (HAdVs) in general and the serotype5 (HAdV-5) in particular are amongst the most popular viral vectors in research and clinical application. However, efficient transduction using HAdV-5 is predominantly achieved in coxsackie and adenovirus receptor (CAR)-positive cells. In the present study, we used the transduction enhancer LentiBOOST^®^ comprising the polycationic Polybrene to overcome these limitations. Using LentiBOOST^®^/Polybrene, we yielded transduction rates higher than 50% in murine bone marrow-derived dendritic cells (BMDCs), while maintaining their cytokine expression profile and their capability to induce T-cell proliferation. In human dendritic cells (DCs), we increased the transduction rate from 22% in immature (i)DCs or 43% in mature (m)DCs to more than 80%, without inducing cytotoxicity. While expression of specific maturation markers was slightly upregulated using LentiBOOST^®^/Polybrene on iDCs, no effect on mDC phenotype or function was observed. Moreover, we achieved efficient HAdV5 transduction also in human monocytes and were able to subsequently differentiate them into proper iDCs and functional mDCs. In summary, we introduce LentiBOOST^®^ comprising Polybrene as a highly potent adenoviral transduction agent for new in-vitro applications in a set of different immune cells in both mice and humans.

## 1. Introduction

A variety of gene delivery techniques have been established over the last decades, which can be subdivided into non-viral and viral strategies. The non-viral subclass comprises methods such assuch as the injection of naked DNA, electroporation, gene gun, sonoporation, magnetofection, and lipoplexes, while for the viral subclass lentivirus (LV), herpes simplex virus (HSV), baculovirus, adeno-associated virus (AAV), and human adenovirus (HAdV) have been used for gene transfer [1]. To date, recombinant HAdVs are the most widely used viral vectors for gene therapy, accounting for 18.6% of vectors used in gene and vaccine therapy clinical trials, especially in the context of different cancer types, HIV, tuberculosis, Ebola virus, malaria, influenza, and, lately, SARS-CoV-2 [1,2,3,4]. In contrast to most other viral vaccines, HAdVs trigger both a humoral response as well as a robust cytotoxic T-cell response, which is favorable with regard to the destruction of virus-infected cells, intracellular pathogens, and cancerous cells [5,6,7].

The best studied member of the HAdV species is serotype 5 (HAdV-5, species HAdV-C) [8]. HAdV5 is characterized by an icosahedral capsid (~90 nm in diameter) and a ~35-kb-long double-stranded linear single DNA genome [9,10]. The viral capsid is mainly comprised of three proteins called hexon, penton base, and fiber, which interact directly and are held together by cement proteins [11,12]. The primary entry receptor for HAdV-5 is the coxsackievirus and adenovirus receptor (CAR) but other receptors are also utilized by HAdVs, such as vascular cell adhesion molecule (VCAM)-1; heparan sulfate proteoglycan (HSPG), major histocompatibility complex (MHC1)-A2, or scavenger receptor A (SR-A) [13]. Hence, HAdV-5s can infect a broad range of dividing and non-dividing cells, including immune cells. Therefore HAdV-5 is not only a very interesting tool to target immune cells in vitro and in vivo for vaccine approaches, but also for gene transfer into cells that are difficult to transfect or electroporate in vitro.

Human monocyte-derived dendritic cells (DCs), which cannot be efficiently transfected using standard transfection reagents or rapidly die after electroporation with DNA, can be transduced with HAdV-5 vectors in vitro to e.g., analyze cell type-specific functions or for genome editing using CRISPR/Cas9 [14,15]. Other approaches aim to modify the function of human DCs for the development of new vaccines by overexpressing therapeutic transgenes. However, high titers of HAdV-5 vectors are often needed to obtain optimal transduction rates in those cells, which are typically associated with higher mortality. Multiplicities of infection (MOI) between 500 and 5000 viral particles per cell are often necessary to achieve transduction rates above 90% in human DCs [16,17,18]. Moreover, transduction efficacy using HAdV-5 vectors for other primary cell types, such as human monocytes or immature murine bone-marrow derived dendritic cells (BMDCs), is often not more than ten percent, even at high titers. The reason for the latter is that HAdV-5 is species-specific and naturally does not infect cells of murine origin [19]. Transduction enhancers are able to overcome these difficulties and a few options are available and tested for adenoviral transduction of different tissues [20,21]. However, given the high sensitivity of DCs towards external stimuli, transduction enhancers with low intrinsic immunogenicity could act as a helpful tool for adenoviral transduction.

In the light of these barriers, we came up with a transduction enhancer developed by Sirion Biotech, the LentiBOOST^®^ reagent. LentiBOOST^®^ is a highly effective, non-cytotoxic transduction enhancer, originally developed for preclinical and clinical application of lentiviral vectors. Two different formulations, optimized for either in-vivo or in-vitro applications, are available. For in-vivo transduction, LentiBOOST^®^ consists ofthe receptor-independently acting Poloxamer 338, developed by Sirion Biotech for clinical use and available in GMP-quality. The Poloxamer 388 has been shown to mediate high transduction rates for clinically relevant cell types, including human CD34+ hematopoietic stem cells (HSCs), primary T cells and NK cells, with LVs [22,23]. Currently, LentiBOOST^®^ is in several clinical trials in the US and Europe up to phase 3, and has been proven to be safe and clinically effective. In 2019, the National Institute of Allergy and Infectious Diseases (NIAID), part of the National Institutes of Health, applied this technology for its lentivirus-based early clinical development of its SCID-X1 trial and was able to develop an enhanced protocol for engineering human CD34+ stem cells [24,25]. For the second formulation, intended for in-vitro experiments, the Poloxamer 388 (LentiBOOST^®^) is combined with Polybrene, a polycationic substance that is used to enhance LV transduction efficiency. Due to toxicity, Polybrene is not suitable for in-vivo applications, but both substances act synergistically on different pathways and in distinctive mode-of actions in enhancing LV transduction efficiency ex vivo. Polybrene reduces the charge–repulsion between the vector and the cell surface by interacting with the negatively charged cellular membrane surface [26], while poloxamers interact with the lipid bilayer and change their physiochemical properties. The effect of poloxamers is depending on the size of the hydrophobic PPO core domain and the temperature and concentration. Studies using supported lipid bilayers (SLBs) on a microcantilever surface confirm an interaction of the poloxamers F68 and F98 with the linear lipid bilayer, resulting in a stronger association of F98, which has a longer PPO core unit compared with F68, with the SLB. Moreover, the poloxamers inhibited lipid diffusion [27].

In the present study, we evaluated the ability of LentiBOOST^®^ mixed with Polybrene to enhance and optimize the transduction efficacy of murine immature (i)BMDCs and mature (m)BMDCs, as well as of human monocytes and immature (i)DCs and mature (m)DCs, with HAdV-5. For this purpose, we first assessed the optimal transduction conditions, with respect to the used multiplicity of infection (MOI) of the HAdV-5 vector, encoding for the green fluorescent protein (GFP), in combination with LentiBOOST^®^/Polybrene, for iBMDCs, mBMDCs, iDCs and mDCs. Using this optimized transduction protocol, we analyzed transduced murine and human DCs further, regarding their survival, expression of typical cell surface markers, secretion of cytokines and their ability to prime naïve allogeneic T cells. Subsequently, we applied our HAdV-5 vector in combination with LentiBOOST^®^/Polybrene also to human monocytes, where we observed high transduction rates. Notably, subsequently we were still able to differentiate these transduced monocytes into iDCs and mDCs, displaying a typical phenotype and function, since these generated mDCs were not only able to prime naïve T cells in a mixed lymphocyte reaction (MLR), but could also be electroporated with a mRNA coding for a tumor antigen, or loaded with peptide to induce a tumor antigen-specific T-cell response in vitro.

## 2. Materials and Methods

### 2.1. Mice

C57BL/6 and BALB/c mice were purchased from Charles River/Wiga (Sulzfeld, Germany) and maintained under pathogen-free conditions according to the European Communities Council Directive (86/609/EEC).

### 2.2. Generation of Murine Bone Marrow-Derived DCs

Bone marrow-derived DCs from C57BL/6 mice were generated as described previously [28] and finally resuspended at a density of 2 × 10^6^ cells in 10-mL R10 medium consisting of RPMI1640, 1% penicillin/streptomycin/L-glutamine, 2-ME and 10% heat-inactivated FBS (GE Healthcare, Chicago, Illinois, United States), additionally supplemented with GM-CSF supernatant (1:10) from a cell line stably transfected with the murine GM-CSF [29]. At days 3 and 6, 10 mL of fresh R10 supplemented with GM–CSF supernatant (1:10) was added, with removing 50% of the old cell culture supernatant at day 6 before. Maturation of BMDCs was induced at day 7 by the addition of 0.1 ng/mL LPS for 20 h. At day 8, cells were used for further experiments.

### 2.3. Generation of Human Monocytes, Monocyte-Derived DCs and T Cells

Human peripheral blood mononucleated cells (PBMCs) of healthy donors were isolated from leucocyte reduction system chambers (LRSCs) using density centrifugation as described previously [30]. For the generation of monocytes, cells were seeded in DC medium consisting of RPMI1640 + 1% penicillin/streptomycin/glutamate + 1% heat-inactivated AB serum + 1% HEPES after removing the non-adherent fraction (NAF). NAF was cryopreserved and stored at −80 °C for isolation of T cells for allogeneic MLR or for antigen-specific T-cell priming.

For the generation of DCs from PBMCs, the adherent cell fraction was cultured for 4 days in DC medium supplemented with 800 IU/mL (day 0) or 400 IU/mL (day 3) recombinant human granulocyte macrophage colony-stimulating factor (GM-CSF) and 250 IU/mL (day 0 and 3) recombinant IL-4 (both Miltenyi Biotec, Bergisch Gladbach, Germany). On day 4, maturation of DCs was induced when indicated by adding a maturation cocktail consisting of 1000 IU/mL IL-6 (Miltenyi Biotec), 200 IU/mL IL-1β (Cell Genix, Freiburg, Germany), 10 ng/mL tumor necrosis factor α (TNF-α; Peptrotech, Hamburg, Germany), and 1 µg/mL prostaglandin E2 (PGE2; Santa Cruz, Dallas, TX, USA) for 24 h before cells were transduced with adenovirus.

### 2.4. Recombinant Adenoviruses

HAdV-5-Luc1 and HAdV-5-GFP are first-generation clones, E1- and E3-deleted replication-deficient adenoviral vectors. HAdV-5-Luc1 contains a CMV-firefly luciferase cassette and was kindly provided by D. T. Curiel from the Washington University School of Medicine, MO, USA. HAdV-5-GFP was cloned as follows: a gene cassette containing a CMV–GFP sequence was inserted into pShuttle. Virus genomes were obtained by homologous recombination of the corresponding shuttle plasmids indicated above with pAdEasy-1 in E. coli BJ5183, as described before [31]. Adenovirus particles were produced by transfection of the different PacI-digested pAd vectors into 293 cells using Lipofectamine 2000. All viruses were amplified in 293 cells and purified by two rounds of CsCl density gradient ultracentrifugation. Verification of viral genomes and exclusion of wild-type contamination were performed by PCR. Physical particle concentration (viral particles (vp)/mL) was determined by OD260 reading. Therefore, a serial dilution of virus stock and viral lysis buffer consisting of 10 mM TE and 0.5% SDS was prepared and incubated at 56 °C for 10 min. To determine the OD260, the dilution factor was multiplied with the subtraction of the absorbance at 320 nm from the absorbance of 260 nm. The mean of three consecutive absorptions was then used as OD260. In addition, Infectious particle concentration was determined by TCID50 assay on 293 cells. Measurement of particle concentration for the HAdV-5-Luc1 vector resulted in an OD of 2.06 × 10^12^ vp/mL and an TCID50/mL of 3.5 × 10^11^, leading to a ratio of 5.9. Concentration of the HAdV-5-GFP vector was identified as 1.22 × 10^12^ vp/mL using OD260 and 7.1 × 10^10^ TCID50/mL resulting in a ratio of 17.2. For experiments, TCID50/mL was used as a basis to calculate the multiplicity of infection.

### 2.5. Adenoviral Transduction

Human monocytes were transduced in either 450 µL (six-well plate)/3 mL (cell culture dish) DC medium (for monocytes) or DC medium supplemented with either 1600 U/mL GM-CSF and 500 U/mL IL-4 (for iDCs). Virus suspension was prepared using 0.5 mg/mL LentiBOOST^®^ + 4 µg/mL Polybrene (Sirion Biotech) or buffer (PBS) only, according to the manufacturer’s instructions. Adenovirus was added to the cells at a MOI of 100 or 200 in a volume of 450 µL (six-well plate) or 3 mL (cell culture dish) resulting in a final infection volume of 900 µL or 6 mL, respectively. After 1.5 h of incubation at room temperature on a rocker and 2.5 h at 37 °C/5% CO_2_, 3 mL (six-well)/18 mL (cell culture dish) of DC medium only (for monocytes) or replenished with cytokines (for iDCs) as described before was added per well. Transduced monocytes were either analyzed by flow cytometry 24 h post infection or were differentiated to day 4 iDCs or day 5 mDCs, as described before, and then used for further experiments.

For adenoviral transduction of human DCs, 5 × 10^5^ iDCs or mDCs were seeded into a 24-well plate in a volume of 125 µL DC medium supplemented with either 1600 U/mL GM-CSF and 500 U/mL IL-4 (for iDCs) or the maturation cocktail consisting of 400 U/mL IL-1𝛽, 2000 U/mL IL-6, 20 ng/mL TNF-𝛼, and 2 µg/mL PGE2, in addition to IL-4 and GM CSF (for mDCs). Virus suspension was prepared as described before. Virus dilutions were performed in DC medium without additives at different MOIs and added to the cells to a final volume of 250 µL. Afterwards cells were incubated as described above, before 1.5 mL of DC medium replenished with cytokines, was added. Cells were used for further experiments 48 h after transduction.

Immature and LPS-matured murine BMDCs were seeded at a density of 5 × 10^5^ cells/well in a 24-well plate in 125 µL of R10 medium. Virus dilutions at different MOIs from 40 to 500 were prepared as described above, using 1 mg/mL LentiBOOST^®^ and 8 µg/mL Polybrene or PBS only, as described by the manufacturer. After adding 125 µL/well virus suspension, BMDCs were incubated as described above, before 1.5 mL of R10 medium was added. Further analyses of transduced BMDCs were performed 48 h afterwards.

### 2.6. Flow Cytometry

Analysis of cells labelled with monoclonal antibodies (mAbs; all BioLegend unless specified otherwise) listed below, and 7AAD was performed using a Canto II flow cytometer (BD Bioscience) and FCS Express 5.1 (DeNovo Software). Anti-human mAbs: CD3 (APC, clone UCHT1), CD14 (PE-Cy7, clone 63D3), CD14 (BV510, clone M5E2), CD16 (PE, clone 3G8), CD19 (APC, BD Bioscience, clone HIB19), CD25 (PE-Cy7, clone BC96), CD56 (APC, clone HIB19), CD80 (APC, clone L307.4), CD83 (PE, clone HB15e), CD86 (BV421, clone IT2.2), HLA-DR (APC-Fire, clone L243). Human DCs were defined as 7AAD^−^, CD11c^+^ and HLA-DR^+^ cells, while monocytes were gated as 7AAD^−^, CD3^−^ and HLA-DR^+^ cells. Anti-mouse mAbs: CD11c (APC, clone N418), CD25 (APC-Cy7, clone PC61), CD80 (BV-421, clone 16-10A1), CD83 (PE, clone Michel-19), CD86 (PE-Cy7, clone GL1), I-A/I-E (BV510, clone M5/114.15.2), HLA-A0201-PE ELAGIGILTV tetramer (produced in house according to Rodenko et al. [32]). Gating on 7AAD^−^, CD11c^+^ cells revealed the population of murine DCs.

### 2.7. Cell Imaging

Cell imaging was performed using the EVOS FL Fluorescence Microscope according to the manufacturer’s instructions.

### 2.8. Dimethylthiazol–Diphenyltetrazolium (MTT) Assay

To perform MTT assays, cells were harvested 48 h after adenoviral transduction. Afterwards, 2 × 10^4^ cells per well were seeded in a 96-well flat-bottom plate. Murine BMDCs were cultivated in 100 µL of R10 medium. Human monocytes and monocyte-derived DCs were cultivated in 100 µL DC medium or DC medium complemented with IL-4 and GM-CSF, respectively, as described before. Cells were incubated for 5 h after adding 10 µL of MTT solution (10 µg/µL in PBS) per well. Formazan crystals were solubilized by adding 100 µL 10% SDS in 0.01 N HCl overnight. Absorbance was measured in duplicates at 570 nm using a Wallac Victor 2 1420 Multilabel Counter (Perkin Elmer, Waltham, MA, USA).

### 2.9. Mixed Lymphocyte Reaction (MLR)

For MLR assays using human matured DCs, cells were co-cultured at different ratios with 2 × 10^6^ allogeneic human T cells derived from NAF. Co-cultures were incubated in 96-well flat bottom cell culture plates for 72 h in 200 μL of DC medium. For murine mature BMDCs, titrated numbers of cells were co-cultured in 96-well flat bottom cell culture plates with 4 × 10^5^ BALB/c derived spleen cells for 72 h in 200 µL of R10 medium. As a positive control, human T cells or murine splenic cells only were incubated together with CD3/CD28 Dynabeads in 200 µL of corresponding cell culture medium, whereas NAF only was regarded as negative control. After 3 days, cell-free supernatants of DC-T cell co-cultures were collected and stored at −20 °C for further analyses. Cells were pulsed with 3H-thymidine (1 μC/well; PerkinElmer) for 16–20 h to determine T-cell proliferation. Thymidine incorporation was measured using a Wallac 1420 Victor2 Microplate Reader (PerkinElmer).

### 2.10. Cytometric Bead Array

Cell culture supernatants were analyzed using the LEGENDplexTM Human Inflammation Panel 1, or -HU Th Cytokine Panel, or -Mouse Inflammation panel, or -MU Th Cytokine Panel (all BioLegend) according to the manufacturer’s instructions.

### 2.11. In-Vitro RNA Transcription and Electroporation of Human DCs

In-vitro transcription of mRNA was performed using the mMESSAGE mMACHINE™ T7 ULTRA Transcription Kit and purified with an RNeasy Kit, according to the manufacturers’ protocols. DCs derived from HLA-A2 positive healthy donors were electroporated with 5 µg MelanA mRNA, as described before [33]. As a control, mDCs were electroporated without mRNA as described by Gerer et al. [33].

### 2.12. Priming of CD8+ T Cells

For priming of antigen-specific T cells as described in [34], DCs were transfected as described above with MelanA mRNA, or were peptide pulsed with the MelanA-derived HLA-A2-binding WT peptide EAAGIGILTV. Thawed autologous NAF was co-cultured at a ratio of 1:10 (final concentration 2 × 10^5^ DCs/mL and 2 × 10^6^ NAF/mL), in MLPC medium consisting of RPMI 1640, 10% human serum, 2 mM L-glutamine, 20 mg/L gentamycin, 10 mM HEPES, 1 mM sodium pyruvate and 1%MEM nonessential amino acids. Fresh MLPC medium was added when necessary. On days 2 and 4, 1000 IU/mL IL-2 (Peprotech) and 10 ng/mL IL-7 were supplemented. Cells were harvested after one week of co-culture and stained with an HLA-A0201-PE ELAGIGILTV tetramer, before adding anti-CD3-APC-H7 and anti-CD8-PE-Cy7. Finally, cells were acquired on a FACS Fortessa (BD Bioscience). Resulting data were analyzed using FCS Express 5.1. as follows: Lymphocytes were gated on by forward scatter (FSC)/sideward scatter (SSC), doublets were excluded by FSC-area and FSC-height, 7-AAD-positive cells were gated out, CD8+ cells were gated on, and CD3 vs. HLA-A tetramer is shown.

### 2.13. Statistical Analysis

Statistical calculations were performed using GraphPad Prism 8 software.

## 3. Results

### 3.1. LentiBOOST^®^ in Combination with Polybrene Enables Transduction of Murine BMDCs with HAdV-5

To overcome the low transduction efficacy due to the lack of CAR expression on target cells, or the species-specificity of the HAdV serotype, we examined the transduction agent LentiBOOST^®^ combined with Polybrene in the context of HAdV transduction, of human and murine primary cells. Therefore, we first transduced murine immature and mature BMDCs with a HAdV-5 coding for GFP at MOIs from 40 to 500, either in the presence or absence of LentiBOOST^®^/Polybrene (LeB/PB; Figure 1). As a negative control, cells were either mock-transduced or transduced with a HAdV-5, encoding for the Renilla firefly luciferase (HAdV-5-Luc1). As assessed by flow cytometry 48 h afterwards, percentage of GFP+ iBMDCs increased from 2% to 9% (MOI40), 3% to 27% (MOI100), 5% to 42% (MOI200) and 11% to 54% (MOI500), using LeB/PB (Figure 1a). Complementing higher transduction rates signal intensity was also increased by using LeB/PB (see Appendix A online). No significant cytotoxicity was observed, except for the highest MOI used in combination with LeB/PB (Figure 1b) and metabolic activity was unaltered (Appendix A online). Similar results were obtained using mBMDCs (Figure 1c,d). Here, the percentage and signal intensity of GFP+ cells were enhanced by the combination LeB/PB with increasing MOIs, from 3% to 15% (MOI40), and from 24% to 54% (MOI500), while survival of mBMDCs was unaltered (Figure 1d and Appendix A online). Viability assessed via an MTT assay revealed a reduced metabolic activity in combining LeB/PB with an MOI500 (Appendix A online). Moreover, LeB/PB did not affect the morphology of either iBMDCs or mBMDCs (Figure 1e,f and Appendix A online).

In summary, LentiBOOST^®^ combined with Polybrene breaks the species-specific HAdV-5 infection barriers of murine immature and mature BMDCs without toxic side effects. As MOIs 200 and 500 of HAdV-5-GFP were shown to be most efficient to transduce immature and mature BMDCs, these were used for the following experiments.

### 3.2. LentiBOOST^®^/Polybrene Does Not Alter the Phenotype and Function of Murine BMDCs

As DCs rapidly react on environmental changes, we next investigated the influence of the LentiBOOST^®^/Polybrene regarding the phenotype and function of immature and mature BMDCs, 48 h after transduction with HAdV-5-Luc1 (MOI500) and HAdV-5-GFP (MOI200 and MOI500), in comparison to cells inoculated with adenovirus alone. Flow cytometric analyses revealed no differences regarding the expression of classical DC cell surface markers, such as CD25, CD80, CD83, CD86 and MHC class II, on iBMDCs (Figure 2a), compared with cells transduced in the absence of LentiBOOST^®^/Polybrene (−LeB/PB). Interestingly, the usage of a GFP encoding adenovirus induced a strong maturation of iBMDCs, independent of the used MOI, which was not observed for firefly luciferase-encoding HAdV-5 (HAdV-5-Luc1) transduced cells, or mock-treated cells (Figure 2a). Expression of CD25, CD80, CD83 and CD86, on the cell surface of mature BMDCs, was not affected by LentiBOOST^®^/Polybrene (Figure 2b). Induction of BMDCs maturation was accompanied by the secretion of pro-inflammatory cytokines, such as MCP-1, TNF-α, IL-6 and IFN-β, while mock- or HAdV-5-Luc1 treated iBMDCS produced only very low, or almost no cytokines (see Appendix A online). Additionally, HAdV-5-GFP +LeB/PB led to an increased production of IL-6 and IFN-β by iBMDCs, when compared with HAdV-5-GFP–LeB/PB (see Appendix A online). A slightly altered secretion of MCP-1, IL-1α and IL-6 was also observed for mBMDCs (see Appendix A online).

To address the functionality of these differentially treated mBMDCs, mixed lymphocyte reactions (MLR) were performed (Figure 3a,b). Here, LentiBOOST^®^/Polybrene (“Mock” + LeB/PB) or in combination with HAdV-5-Luc1 (MOI 200 and MOI 500), did not affect the capability of mBMDCs to prime allogeneic T cells. By contrast, transduction of mBMDCs with HAdV-5-GFP in the presence of LentiBOOST^®^/Polybrene led to a stronger proliferation of T cells at a DC:T cell ratio of 1:10 (1.3 fold MOI200/1.7 fold MOI500) and 1:33 (1.5 fold MOI200/1.7 fold MOI500). This effect was not observed for HAdV-5-GFP transduced BMDCs missing the LeB/PB. Next, we examined the portfolio of classical T-cell-derived cytokines in the supernatants of the co-cultures. We observed no differences between BMDCs transduced in the absence (−LeB/PB) or presence (+LeB/PB), to induce classical Th1/Th2 associated cytokines IL-2 and IL-6 (Figure 3c), Th1 cytokines IFN-γ and TNF-α (Figure 3d), Th2 cytokines IL-4 and IL-13 (Figure 3e), or Th17 cytokines IL-17A and IL-22 (Figure 3f).

Taken together, phenotype and function of BMDCs are not altered by LentiBOOST^®^/Polybrene, although a slight increase (i) in the secretion of pro-inflammatory cytokines by iBMDCs, and (ii) the capability of mBMDCs to prime T cells was observed, partially depending on the transgene encoded by the adenovirus.

### 3.3. LentiBOOST^®^/Polybrene Increases Transduction Efficiency of Human Monocyte-Derived DCs at Low HAdV-5 Virus Titers

Although lacking CAR expression, human DCs can be transduced with HAdV-5 at high titers >MOI500. Since high virus titers are accompanied by toxicity, followed by loss of function, we next assessed if LentiBOOST^®^/Polybrene can improve HAdV-5-mediated gene delivery to human DCs. Flow cytometric analyses, 48 h after transduction of immature (i) DCs and mature (m)DCs with HAdV-5-GFP at different MOIs, revealed that a MOI of 10 is already sufficient to infect ~60% of iDCs, while without LentiBOOST^®^/Polybrene only 1% of cells were GFP-positive (Figure 4a). Using a MOI of 100 or 200, >80% of cells were GFP-positive, while only 22% or 42% of cells were transduced in the absence of LentiBOOST^®^/Polybrene, respectively. Toxicity was moderate, with a decrease of living cells from 96% to 80% (MOI100), or 92% to 65% (MOI200; Figure 4b). The transduction efficacy of mDCs increased from 1% to 20% (MOI 10), 1% to 49% (MOI 20), 13% to 67% (MOI40), 43% to 84% (MOI100) and from 63% to 84% (MOI200), using the LeB/PB (Figure 4c). Using LentiBOOST^®^/Polybrene for transduction highly increased signal intensity (see Appendix A online). Again, survival of mDCs was not impaired (Figure 4d) and viability remained unaltered (see Appendix A online). Moreover, iDCs and mDCs showed a similar morphology and cell clustering may be increased in the presence of LentiBOOST^®^/Polybrene, independent of the adenovirus (Figure 4e,f, see Appendix A online). Overall, LentiBOOST^®^ combined with Polybrene efficiently transduces iDCs and mDCs, using a MOI of 10 or 40, respectively, with an optimal MOI of 100.

### 3.4. LentiBOOST^®^/Polybrene Induces Maturation of Immature Human DCs but Does Not Alter DC Function

Next, we analyzed if LentiBOOST^®^/Polybrene and the observed enhanced infection rate influence the phenotype and function of human DCs. Thus, iDCs and mDCs, either mock-treated, or transduced with HAdV-5-Luc1 (MOI200), or HAdV-5-GFP (MOI 100 and 200), were analyzed by flow cytometry, 48 h post infection. In contrast to murine iBMDCs, human iDCs exposed to LentiBOOST^®^/Polybrene showed a significant upregulation of maturation markers CD83 and CD86, while CD25, CD80 and HLA-DR were not statistically altered (Figure 5a). For HAdV-5-GFP the median was even higher when compared with mock- or HAdV-5-Luc1 controls. Interestingly, we observed almost no differences regarding the cytokine production, except for IL-8, which was clearly upregulated in the presence of the LentiBOOST^®^/Polybrene (see Appendix A online). However, inoculation of mDCs with LentiBOOST^®^/Polybrene did not alter their expression profiles in comparison to control mDCs (Figure 5b), nor changed their cytokine production (see Appendix A online). Moreover, median fluorescence intensity was strongly increased for all analyzed cell surface markers, including CD80, CD83, CD86 and HLA-DR, when compared with LentiBOOST^®^/Polybrene treated iDCs.

Next, to analyze the mDCs-mediated allogeneic T-cell proliferation, MLR assays were performed. Results revealed that neither LentiBOOST^®^/Polybrene nor the amount of adenovirus used influenced the T-cell stimulatory capacity of mDCs (Figure 6a,b). When analyzing the supernatants derived from the DC-T-cell co-cultures for their content of classical T-cell-derived cytokines, such as IL-2, IL-6, IFN-γ, IL-5, IL-13 and IL-22, no differences between +LeB/PB and –LeB/PB were observed (Figure 6c–f). However, co-cultures of T cells with HAdV-5-GFP-transduced mDCs showed reduced levels of IL-5, IL-6, IL-13 and IL-22, independent of the LentiBOOST^®^/Polybrene.

Taken together, HAdV-5-mediated gene transfer, in the presence of LentiBOOST^®^/Polybrene, does not significantly interfere with the DC phenotype and their function.

### 3.5. Efficient HAdV-5-Mediated Gene Transfer into Monocytes by LentiBOOST^®^/Polybrene Allows for Proper Subsequent iDCs and mDCs Differentiation

Similar to murine BMDCs, human monocytes are poorly permissive to HAdV-5, even at high titers. Hence, we transduced freshly isolated human monocytes (see Appendix A online) with HAdV-5-GFP at a MOI of 100 or 200, either in the absence or presence of LentiBOOST^®^/Polybrene. Twenty-four hours afterwards, cells were analyzed by flow cytometry. As depicted in Figure 7a, the transduction efficacy increased from 7% to 36% (MOI100) and 15% to 36% (MOI200), when applying LentiBOOST^®^/Polybrene to the cell culture. Notably, mean GFP expression was upregulated from 227 to 3591 (MOI100) and 588 to 4264 (MOI200) (see Appendix A online), while no toxicity (Figure 7a), no changes in the composition of monocyte subpopulations (see Appendix A online), no changes in metabolic activity (see Appendix A online) or cytokine secretion (see Appendix A online), were observed. Next, transduced monocytes were differentiated into iDCs and mDCs and analyzed regarding their phenotype and function. Flow cytometric analyses of day four iDCs, differentiated from transduced monocytes, using IL-4 and GM-CSF, revealed no loss of GFP positivity, in comparison to monocytes, although median values were diminished (Figure 7b). Again, survival of cells was shown to be >92% (Figure 7b) and cell viability remained unaffected (see Appendix A online). Further exploration of these cells displayed a typical immature DC-phenotype regarding the expression of CD25, CD80, CD83, CD86 and HLA-DR, in comparison to the “Mock”-control, independent of the presence or absence of LentiBOOST^®^/Polybrene (Figure 7c). In addition, cytokine profiles were unaltered (see Appendix A online).

Next, mDCs were generated from these iDCs by adding a maturation cocktail, consisting of IL-1β, IL-6, PGE2 and TNF-α for another 24 h. While GFP expression and survival of those cells was comparable with iDCs shown before (Figure 7d), matured DCs highly upregulated typical DC maturation markers, such as CD25, CD80, CD83, CD86 and HLA-DR (Figure 7e). Importantly, the LentiBOOST^®^/Polybrene did not affect this maturation process. However, exposure of monocytes to HAdV-5-GFP alone led to a 2.6- (MOI100) or 2.4 (MOI200)-fold increased expression of CD25 on mDCs, which was not observed when LeB/PB was present during transduction. Moreover, transduction with HAdV-5-GFP affected upregulation of CD83 in a virus concentration dependent manner, which in contrast to CD25 was independent of the usage of LentiBOOST^®^/Polybrene. Interestingly, in the supernatants of mock-treated cells, without LeB/PB, we found strongly increased IL-6, TNF-α and MCP-1 concentrations (see Appendix A online).

Finally, we investigated the function of these mDCs, using an allogeneic MLR assay and in addition, we used them to prime autologous T cells, in a tumor-antigen-specific manner (Figure 8). Regarding MLR assays, mock- and HAdV-5-GFP-treated monocytes −/+ LeB/PB, differentiated into mDCs were co-cultured with allogeneic T cells at different ratios, before T-cell proliferation was assessed by thymidine incorporation (Figure 8a). Although we observed a HAdV-5-GFP- and virus concentration- mediated effect regarding the capacity of mDCs to prime naïve T cells, no LentiBOOST^®^/Polybrene dependent alterations were observed. In addition, transduced cells (−/+LeB/PB), were still able to induce a high and efficient T-cell proliferation accompanied by an unaltered IL-2 secretion (Figure 8a,b).

To prime autologous T cells, mDCs were additionally electroporated with RNA coding for the tumor antigen MelanA (MelA) or loaded with a MelA-specific peptide. Survival of DCs four hours after electroporation varied from 40% to 60% (Figure 8c). Next, MelA-peptide-loaded, MelA RNA or control electroporated (“no RNA”) DCs, were used to stimulate bulk autologous CD8+ T cells for one week, before the fraction of MelA-specific T cells was determined by tetramer staining. As shown in Figure 8d and Appendix A (see Appendix A online), MelA-electroporated and MelA-peptide-loaded DCs induced antigen-specific CD8+ T cells, compared with control DCs (“no RNA”), even when they had been transduced with HAdV-5 and treated with LentiBOOST^®^/Polybrene before. Noteworthy, the numbers of MelA+ CD8+ T cells vary in a donor-dependent manner (see Appendix A online), which is rather common when cells derived from different healthy donors are used. However, these data clearly indicate that LentiBOOST^®^/Polybrene does not impair the capability of DCs to express, process and present a tumor-specific antigen in an MHC-class I-restricted manner.

In summary, using LentiBOOST^®^/Polybrene not only allows for the highly efficient transduction of human monocytes with HAdV-5, but additionally for the differentiation into iDCs and functional mDCs.

## 4. Discussion

Using directed viral gene transfer to treat human diseases, gene therapy holds the potential to revolutionize medicine [1]. Currently, HAdV vectors attract tremendous attention in the context of newly developed vaccines against SARS-CoV-2. Amongst those that have been already applied and proven safe and highly effective, some are based on a chimpanzee adenovirus (AZD1222, Jenner Institute/AstraZeneca/University of Oxford), or on human adenovirus types 5 and -26 (Ad5-nCoV, Cansino Biologics/Beijing Institute of Biotechnology; JNJ-78436735 [Ad26]; Janssen Pharmaceutical Companies of Johnson and Johnson) [35]. However, usage of HAdV-5 vectors is limited regarding the transduction of cells lacking appropriate receptors on their cell surface (e.g., monocytes) or species-specificity (e.g., murine cells such as BMDCs) [19,36]. In the past, several techniques have been developed to promote virus-mediated gene delivery comprising (i) physical methods, (ii) genetic bioengineering of viruses and (iii) chemical methods, namely material additives [37]. Amongst these, polymers are the most extensively studied delivery systems for viral vectors such as HAdV-19 to increase the efficiency of HAdV-mediated gene transfer into epithelial and endothelial as well as mesenchymal stem cells [38,39]. Besides others, the cationic polymer polybrene is a common enhancer of viral delivery in vitro due to its cost efficiency and its simple and safe handling [37,40]. Poloxamers, on the other hand, are commercially available, FDA-approved, thermoresponsive triblock copolymers, consisting of two blocks of hydrophilic poly(ethylene oxide) (PEO) and one block of hydrophobic poly(propylene oxide) (PPO) [37]. Poloxamer 338 (PEO141-PPO44-PEO141) has been described to efficiently enhance LV delivery in T cells 19. In combination with polybrene, Höfig and colleagues reported a further elevated transduction efficacy, which was explained by the distinct modes of each adjuvant, polybrene-compensating electrostatic repulsion and poloxamer 338 fluidization of the membrane [37,41]. LentiBOOST^®^/Polybrene by Sirion Biotech is such a two-component transduction enhancer and has already been used to efficiently promote LV transduction in vitro to generate melanoma-specific human T cells for cancer immunotherapy [42]. In the present study, we used LentiBOOST^®^/Polybrene to transduce not only murine BMDCs but also human monocyte-derived DCs as well as monocytes efficiently with HAdV-5 in vitro to overcome the barriers of receptor dependency and species specificity. In accordance with previous published data, we report here that, even at high virus titers (MOI 200 and MOI 500), transduction efficacy for iBMDCs reached only approx. 5% to 11%, and 13% and 24%, for mBMDCs, respectively. While alternative HAdV receptors VCAM-1 and SR-A are constitutively expressed by bone marrow-derived cells [43,44], MHC class molecules are upregulated upon maturation of BMDCs, which might contribute to the higher infection rate. Nevertheless, a much higher transduction efficacy is the prerequisite of success for most in-vitro and in-vivo approaches. Using LentiBOOST^®^/Polybrene, transduction efficacy increased to 54% for immature and mature BMDCs, without any serious toxicity, change in phenotype and function of these cells. Interestingly, in comparison to HAdV-5-Luc1 the HAdV vector containing the green fluorescent protein (GFP) induced maturation of iBMDCs independent of the LentiBOOST^®^/Polybrene, while for mBMDCs no differences in the maturation status were observed. Moreover, GFP-transduced cells showed a higher capacity to induce T-cell proliferation. GFP was detected in 1961 as a by-product of the extraction of aequorin from the Auquorea victoria jellyfish [45], and since then it has been developed further and is still widely used to label cells or to track gene expression. Of notice, several papers described an immunogenic effect of GFP, being slightly immunogenic in C57BL/6 mice and exhibiting a higher immunogenicity in BALB/c mice [46,47,48]. However, a later study postulated that when used as an intracellular gene reporter, GFP is processed in the host cell followed by presentation at the cell’s surface via MHC class I molecules, leading to an antigen-specific recognition by cytotoxic T-lymphocyte [48]. Similarly, in human DCs, GFP expression was shown to act as an “adjuvant” to enhance T-cell immunity to the melanoma tumor antigen MART1 and to expand multiple CD4+ and CD8+ T-cell clones [49]. For murine DCs transduced with HAdV-5-LucI, encoding for the luciferase gene, no induction of maturation in iBMDCs was observed, despite Limberis et al. describing an immunogenic response of C57BL/6 mice towards luciferase in vivo [50].

Within in this study, human iDCs transduced with HAdV-5-GFP showed only an enhanced expression of maturation markers CD80, CD83, CD86 and HLA-DR in comparison to control cells, whereas no differences were observed for mDCs. Moreover, neither the transfer of GFP into mDCs, nor the use of LentiBOOST^®^/Polybrene resulted in an altered capability of mDCs to stimulate T cells. Merely T cells co-cultivated with HAdV-5-GFP-transduced mDCs produced less Th2-cytokines IL-5, IL-6 and IL-13, thereby in part supporting the postulated immunogenicity of GFP for human DCs reported by Francesca Re and colleagues [49].

Although Polybrene has been used to improve LV-mediated gene delivery to murine and human DCs, to the best of our knowledge no publication addressed the maturation status of DCs treated with Polybrene only [51,52]. This applies also to the LentiBOOST^®^, the Poloxamer 338, which in this study has been used for the first time in combination with Polybrene to improve HAdV-mediated gene transfer into DCs and monocytes. Notably, LentiBOOST^®^/Polybrene increased HAdV-mediated GFP gene transfer into iDCs approximately four-fold (MOI 100) and two-fold (MOI 200), to more than 80% GFP positive cells. Interestingly, this high infection rate is not accompanied by a maturation signal as confirmed by the mock- and HAdV5Luc1 controls.

In a last set of experiments, we efficiently transduced human monocytes with HAdV-5 followed by proper differentiation of these cells into immature and mature DCs. Usually, Adenoviruses infect human CD14+ monocytes ineffectively, mainly due to the lack of appropriate expression of integrins αvβ3 and αvβ5 [53], which are the receptors for HAdV-5 on these cells [54,55]. Using LentiBOOST^®^/Polybrene in combination with HAdV-5-GFP increased the number of GFP+ monocytes 5.3-fold (MOI 100) and 2.4-fold (MOI 200) to 36.4 percent. Importantly, this GFP expression was stable during differentiation of monocytes to iDCs and mDCs, thereby allowing for a sustained expression of the delivered gene of interest in transduced DCs.

Finally, we analyzed these transduced monocytes differentiated to mDCs with respect to their functionality. Compared with non-transduced cells, these mDCs retained their full capacity to prime allogeneic T cells, independently of the transduction enhancer substance. Notably, when these mDCs were subjected to electroporation with a tumor mRNA encoding MelanA, they endured this stress test and could be used to prime antigen-specific CD8+ T cells. Although mDCs not transduced with HAdV-5-GFP (+/−LeB/PB)—as monocytes usually induced higher numbers of CD3+ CD8+ MelanA+ T cells—efficiently infected (HAdV5GFP + LeB/PB) and electroporated cells were still functional and elicited sufficient antigen-specific T-cell responses in vitro. However, the capability of T-cell priming varied between donors. This might be due to the different frequencies of MelanA-specific naïve precursors in the blood of PBMCs of healthy donors. Moreover, these differences in T-cell reactivity reveal the heterogeneity of the immunological status of the different donors in general.

Taken together, here we present the transduction enhancer LentiBOOST^®^/Polybrene as a promising and valuable tool to increase HAdV transduction efficacy, not only of murine and human DCs but also of human monocytes, without compromising the function of those cells. Therefore, using LentiBOOST^®^/Polybrene shows potential in promoting HAdV- or LV-mediated viral gene transfer in vitro as well as in vivo for future efficient and safe clinical applications.

## Figures and Tables

**Figure 1 viruses-14-00092-f001:**
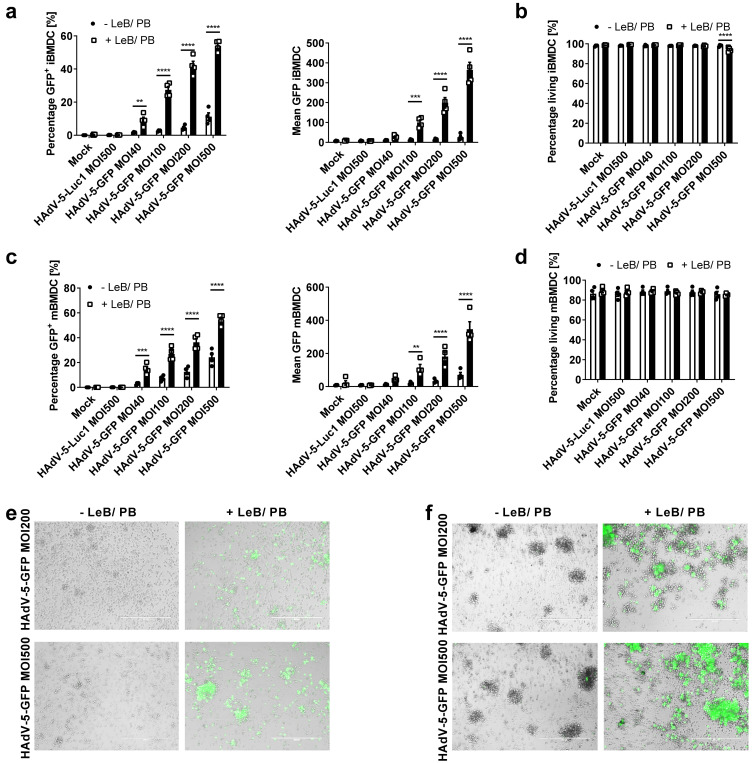
LentiBOOST^®^/Polybrene highly increase transduction efficacy of murine BMDCs. Immature (i) and LPS-matured (m)BMDCs were transduced with HAdV-5-GFP at different MOIs in combination with LentiBOOST^®^/Polybrene (+LeB/PB) or with water (control), for 48 h. As a control, cells were mock-transduced (“Mock”) or transduced using HAdV-5-Luc1. (**a**,**c**) Flow cytometric analyses of iBMDCs (**a**) and mBMDCs. (**c**) GFP expression presented as percentage and mean fluorescence of GFP+ BMDCs. (**b**,**d**) Flow cytometric analyses of iBMDCs (**b**) and mBMDCs (**d**) regarding 7AAD- living cells. (**a**–**d**) Data are mean ± SEM of four independent experiments with cells derived from different mice. Two-way ANOVA and Sidak correction were performed. ** *p* < 0.01, *** *p* < 0.001, **** *p* < 0.0001, bars without annotation are not significant (*p* > 0.05) in comparison to the respective condition water control. Black circles represent conditions without LentiBOOST^®^/Polybrene, white squares represent conditions with LentiBOOST^®^/Polybrene. (**e**,**f**). Fluorescence imaging of HAdV-5-GFP-transduced iBMDCs (**e**) and mBMDCs (**f**) 48 h after transduction. Scale bar (white line) = 400 µm. One representative experiment out of four is shown.

**Figure 2 viruses-14-00092-f002:**
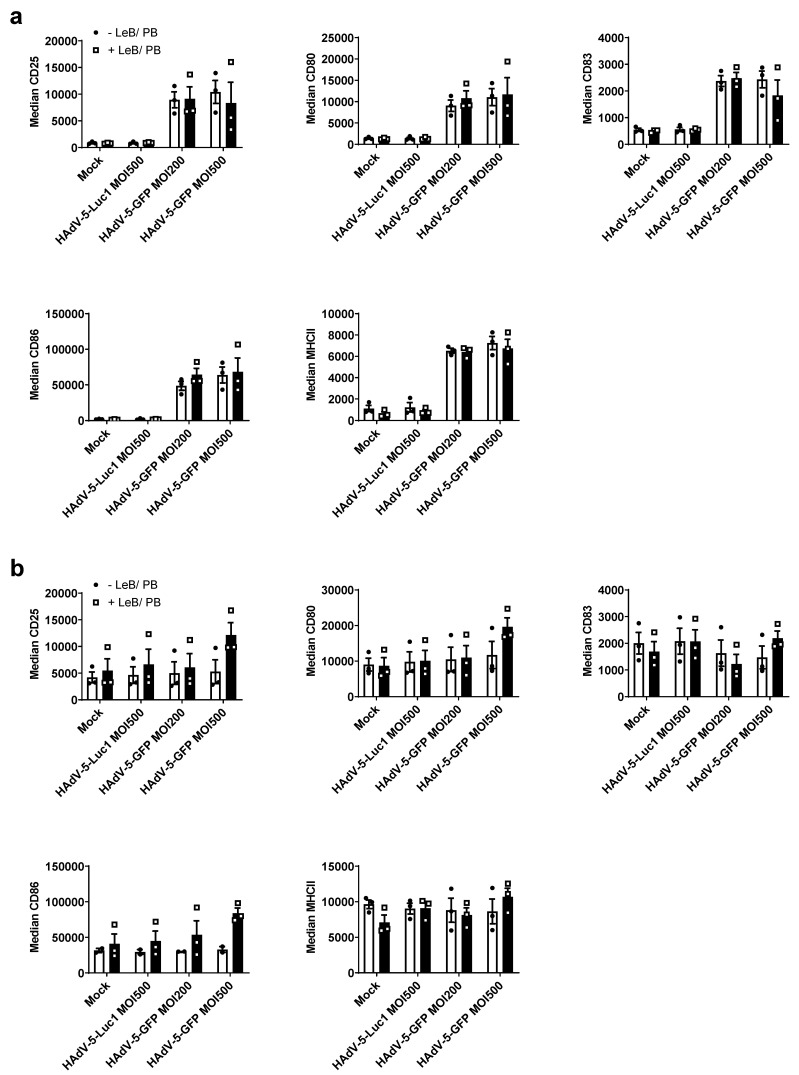
LentiBOOST^®^/Polybrene does not impair the maturation status of BMDCs. (**a**,**b**) Immature and LPS-matured BMDCs were transduced with HAdV-5-Luc1 at a MOI of 500, or with HAdV-5-GFP at a MOI of 200 and 500 in combination with LentiBOOST^®^/Polybrene (+LeB/PB) or a water control, for 48 h. As a negative control, cells were mock transduced (“Mock”). Flow cytometric analyses regarding CD25-, CD80-,CD83, CD86- and MHC-II expression of iBMDCs (**a**) and mBMDCs (**b**), presented as median fluorescent intensity of positive cells. Results are shown as mean ± SEM of three independent experiments with cells derived from different mice. Black circles represent conditions without LentiBOOST^®^/Polybrene, white squares represent conditions with LentiBOOST^®^/Polybrene..Two-way ANOVA and Sidak correction were performed. Bars without annotation are not significant (*p* > 0.05) in comparison to the respective condition without (−)LeB and PB.

**Figure 3 viruses-14-00092-f003:**
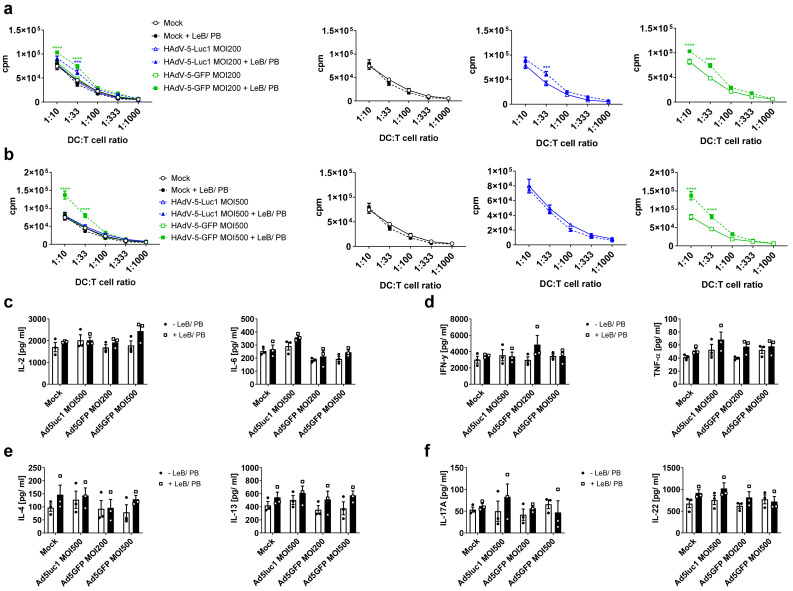
HAdV-5-transduced mBMDCs in combination with LentiBOOST^®^/Polybrene retain their function. LPS-matured BMDCs were transduced with HAdV-5-Luc1 or HAdV-5-GFP at an MOI of 200 or 500 in combination with LentiBOOST^®^/Polybrene (+LeB/PB) or water (−LeB/PB). Forty-eight hours afterwards, cells were assessed using an allogeneic mixed lymphocyte reaction assay. (**a**,**b**) Allogeneic mixed lymphocyte reactions (MLRs) of mBMDCs transduced at MOI200 (**a**) or MOI500 (**b**). Cells were co-cultured with BALB/c derived spleen cells at the indicated ratios for 72 h, before pulsing with 1 µC/well [H3]-thymidine. Thymidine uptake was assessed using a 1450 MicroBeta counter. (**c**–**f**) Cytometric bead array (CBA) of MLR-derived cell culture supernatants. Cell-free supernatants of the DC-T cell co-cultures described in (**a**,**b**), were collected at day 3 before pulsing with thymidine and stored at −20 °C. Content of cytokines was determined by cytometric bead array. (**a**–**f**) Data are mean ± SEM of three independent experiments with cells derived from different mice. Black circles represent conditions without LentiBOOST^®^/Polybrene, white squares represent conditions with LentiBOOST^®^/Polybrene.Two-way ANOVA and Sidak correction were performed. *** *p* < 0.001, **** *p* < 0.0001, bars without annotation are not significant (*p* > 0.05) in comparison to the respective condition –LeB/PB.

**Figure 4 viruses-14-00092-f004:**
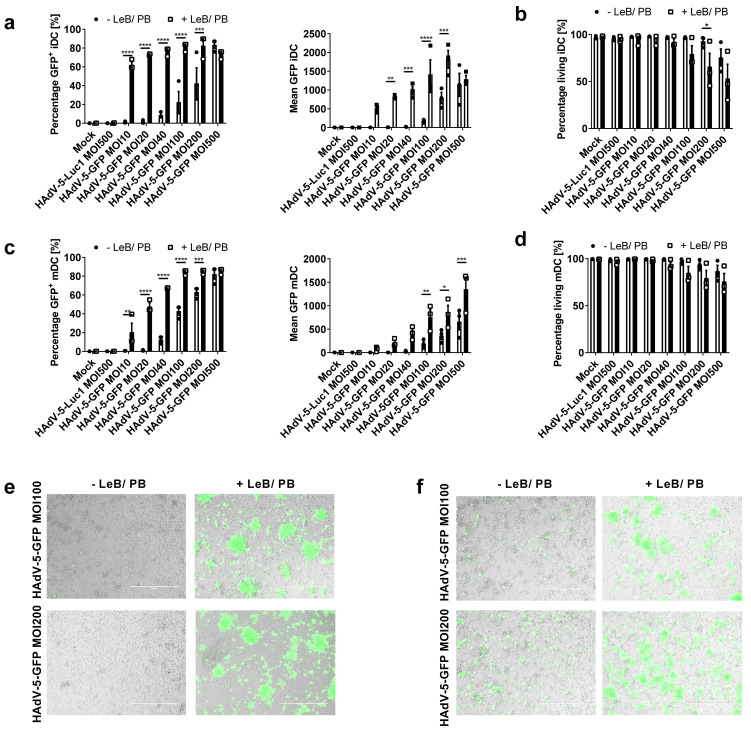
LentiBOOST^®^/Polybrene enables high transduction rates of immature and mature human DCs with HAdV-5. Human monocyte-derived immature (iDC) and mature (m)DCs were transduced with HAdV-5-Luc1 or HAdV-5-GFP at indicated MOIs, ranging from MOI 10 to 50 in combination with LentiBOOST^®^/Polybrene (+LeB/PB) or water only (−LeB/PB). As a negative control, cells were not transduced with adenovirus (“Mock”). Forty-eight hours afterwards, cells were analyzed for expression of GFP and survival. (**a**–**d**) Flow cytometric analyses of iDCs (**a**,**b**) and mDCs (**c**,**d**) regarding their GFP expression (**a**,**c**) depicted as percentage positive cells and mean fluorescent intensity, as well as on 7AAD- living cells (**b**,**d**). Data are mean ± SEM of three independent experiments with cells derived from different healthy donors. Two way ANOVA and Sidak correction were performed. * *p* < 0.05, ** *p* < 0.01, *** *p* < 0.001, **** *p* < 0.0001, bars without annotation are not significant (*p* > 0.05) in comparison with the respective condition–LeB/PB. Black circles represent conditions without LentiBOOST^®^/Polybrene, white squares represent conditions with LentiBOOST^®^/Polybrene. (**e**,**f**) Fluorescence imaging of iDCs (**e**) and mDCs (**f**) two days after transduction with HAdV5GFP. Scale bar (white line) = 400 µm One representative experiment out of three is shown.

**Figure 5 viruses-14-00092-f005:**
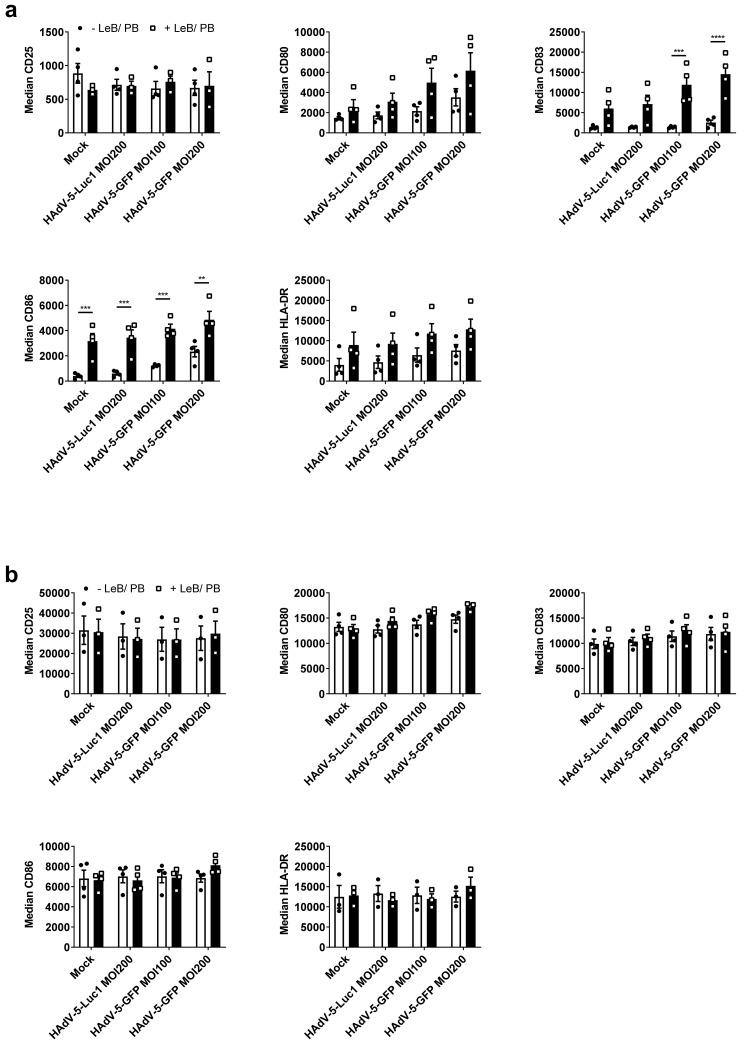
Induction of maturation in immature DCs following adenoviral transduction in the presence of LentiBOOST^®^/Polybrene. (**a**,**b**) Human monocyte-derived iDCs and mDCs were transduced with HAdV-5-Luc1 at a MOI of 200, or with HAdV-5-GFP at a MOI of 100 and 200 in combination with LentiBOOST^®^/Polybrene (+LeB/PB) or water (−LeB/PB), for 48 h. Non-transduced cells served as negative control (“Mock”). Afterwards, median fluorescent intensity of CD25, CD80, CD83, CD86 and HLA-DR was determined by flow cytometry for iDCs (**a**) and mDC (**b**). Data are mean ± SEM of four independent experiments with cells derived from different healthy donors. Black circles represent conditions without LentiBOOST^®^/Polybrene, white squares represent conditions with LentiBOOST^®^/Polybrene. Two-way ANOVA and Tukey correction were used. ** *p* < 0.01, *** *p* < 0.001, **** *p* < 0.0001, bars without annotation are not significant (*p* > 0.05) in comparison to the respective condition −LeB/PB.

**Figure 6 viruses-14-00092-f006:**
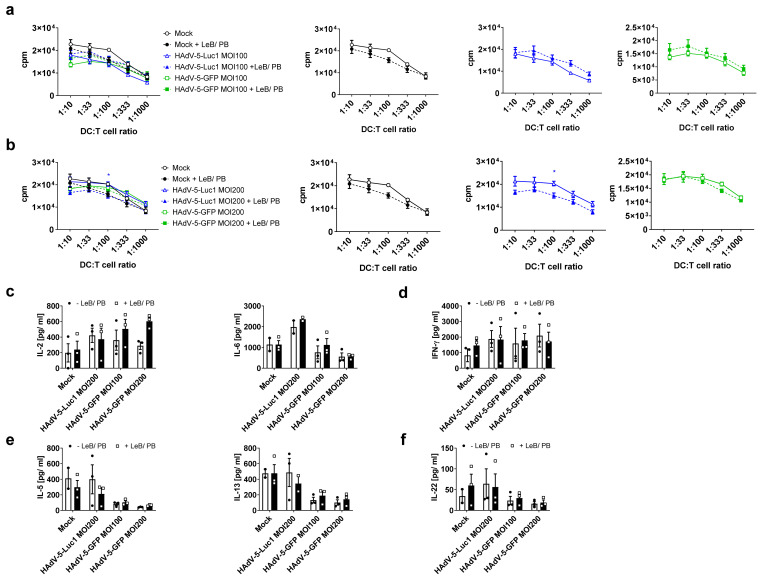
Enhanced adenoviral transduction efficacy using LentiBOOST^®^/Polybrene retains the functionality of human mDCs. Mature dendritic cells were transduced with HAdV-5-Luc1 or HAdV-5-GFP at a MOI of 100 and 200 or PBS (“Mock”) in combination with LentiBOOST^®^/Polybrene (+LeB/PB) or water only (−LeB/PB), for 48 h. (**a**,**b**) Allogeneic MLR assays with mature DCs transduced with HAdV-5-Luc1/HAdV-5-GFP at a MOI of 100 (**a**) and a MOI of 200 (**b**). Mature DCs and T cells, derived from the non-adherent fraction (NAF) of peripheral blood mononuclear cells (PBMCs), were co-cultivated at the indicated ratios for 72 h and subsequently pulsed with 1 µC/well [H3]-thymidine for 16 h. Thymidine uptake was assessed using a 1450 MicroBeta counter. (**c**–**f**) Cell-free supernatants derived from MLR assays described in (**a**,**b**) were analyzed by CBA for their content of cytokines, 72 h after co-culture. (**a**–**f**) Data are mean ± SEM of three independent experiments with cells derived from different healthy donors. Two-way ANOVA and Sidak correction were performed. * *p* < 0.05, bars without annotation are not significant (*p* > 0.05) in comparison to the respective condition–LeB/PB. Black circles represent conditions without LentiBOOST^®^/Polybrene, white squares represent conditions with LentiBOOST^®^/Polybrene.

**Figure 7 viruses-14-00092-f007:**
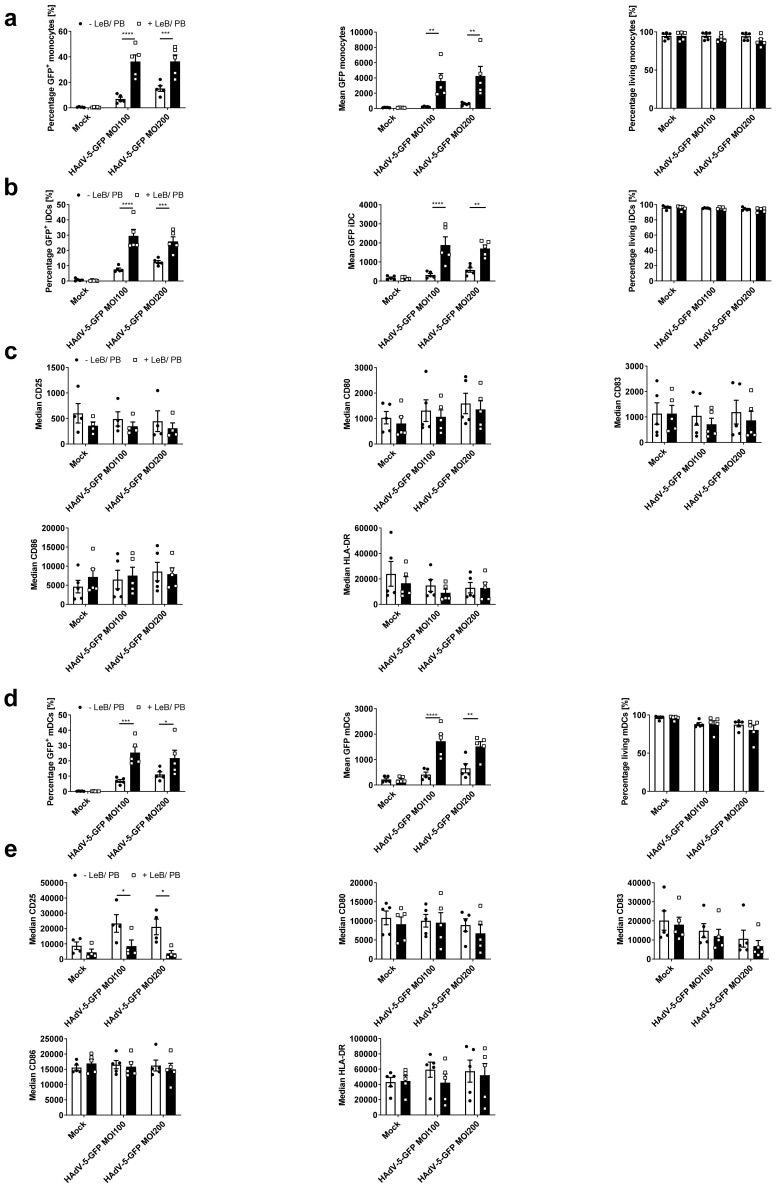
Human monocytes efficiently transduced in the presence of LentiBOOST^®^/Polybrene can be differentiated into iDCs and mDCs. Human monocytes were transduced with HAdV-5-GFP at a MOI of 100 and 200 in combination with LentiBOOST^®^/Polybrene (+LeB/PB) or water only (−LeB/PB). Non-infected cells (“Mock”) served as a negative control. Monocytes were either immediately analyzed by flow cytometry or differentiated into iDCs for 4 days using GM-CSF and IL-4, or into day 5 mDCs, additionally adding IL-1β, IL-6, PGE2 and TNF-α for 24 h. (**a**,**b**,**d**) Flow cytometric analyses on the percentage and mean fluorescent intensity of GFP positive as well as on 7-AAD negative living CD14+ HLA-DR+ monocytes (**a**), CD11c+ HLA-DR+ iDCs (**b**), or CD11c+ HLA-DR+ mDCs (**c**). (**c**,**d**) Flow cytometric analyses on cell surface markers CD25, CD80, CD83, CD86 and HLA-DR on iDCs (**c**) and mDCs (**e**). (**a**–**e**) Data are mean ± SEM of five independent experiments with cells derived from different healthy donors. Black circles represent conditions without LentiBOOST^®^/Polybrene, white squares represent conditions with LentiBOOST^®^/Polybrene. Two-way ANOVA and Sidak correction were performed. * *p* < 0.05, ** *p* < 0.01, *** *p* < 0.001, **** *p* < 0.0001, bars without annotation are not significant (*p* > 0.05) in comparison to the respective condition–LeB/PB.

**Figure 8 viruses-14-00092-f008:**
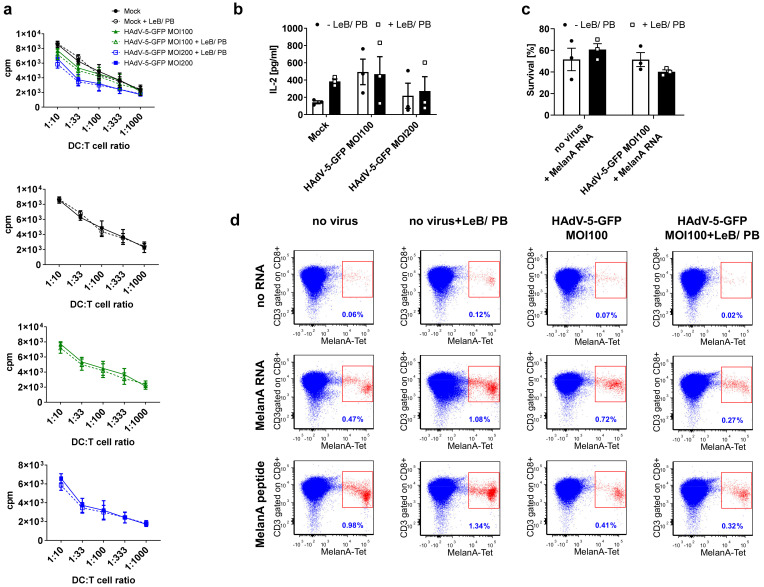
Mature DCs differentiated from transduced monocytes in the presence of LentiBOOST^®^/Polybrene are fully functional. Human monocytes were non-transduced (“Mock” or “non virus”) or transduced with HAdV-5-GFP (MOI100 or MOI200) either in the presence of buffer only (−LeB/PB) or LentiBOOST^®^/Polybrene (+LeB/PB) as described before. Afterwards, cells were differentiated into mDCs. (**a**) Mixed lymphocyte reaction (MLR) assays. Mature DCs and allogeneic T cells were co-cultivated at the indicated ratios for 72 h and subsequently pulsed with 1 µC/well [H3]-thymidine for 16 h. Thymidine uptake was assessed using a 1450 MicroBeta counter. (**b**) Cell-free supernatants derived from MLR assays described in (**a**), were analyzed by CBA for their content of IL-2. (**c**,**d**) Priming of autologous T cells in a tumor-antigen-specific manner. Mature DCs derived from transduced monocytes were either electroporated with RNA coding for MelanA or loaded with a MelanA-derived peptide (EAAGIGILTV), for 1 h. Dendritic cells electroporated without RNA (“no RNA”) served as a control. (**c**) Survival of non-transduced-, or HAdV5GFP transduced mDCs either in the presence or absence of LentiBOOST^®^/Polybrene was assessed before and 4 h after electroporation with MelanA RNA by trypan blue staining. Percentage of living cells after electroporation compared with living cells before electroporation is shown. Data are mean ± SEM of four (**a**,**b**) or three (**c**) independent experiments with cells derived from different healthy donors. Two-way ANOVA and Sidak correction were performed. Bars without annotation are not significant (*p* > 0.05) in comparison to the respective condition–LeB/PB. (**d**) Four hours after electroporation, these DCs were used to stimulate autologous NAF cells at a 1:10 ratio. One week after stimulation, the percentage of MelanA-specific CD3+ CD8+ T cells (depicted in red) was analyzed by tetramer-staining and flow-cytometry. Indicated percentages refer to the total number of CD8+ T cells. Shown here is one representative experiment out of three.

## Data Availability

The data presented in this study are available in the article.

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
