# Peer review of "Breaking Entry-and Species Barriers: LentiBOOST^®^ Plus Polybrene Enhances Transduction Efficacy of Dendritic Cells and Monocytes by Adenovirus 5"

_viruses, 2022, doi:10.3390/v14010092_

Round 1

Reviewer 1 Report

The Authors have sufficiently addressed all the comments from the first round of review. 

Reviewer 2 Report

Dear Dr Knippertz and Strack
Thank you for considering the reviewers' comments and for including new elements in your manuscript. 
The quality of this new version has been significantly increased after responding point by point to the comments.
I would again call your attention to comments 18 and 20.

This manuscript is a resubmission of an earlier submission. The following is a list of the peer review reports and author responses from that submission.

Round 1

Reviewer 1 Report

The authors describe an original method to increase the efficiency of gene transfer in dendritic cells using adenovirus vector. This approach is documented in vitro on both human and murine cells. Transduced primary cells retain their stem cell properties and some characteristics of immune cells. The data presented here are convincing, nevertheless some points need to be clarified.

What are the analytical methods for flow cytometry?

The evidence of non-toxicity is based on the determination of the number of living cells. How is the % of living cells determined? A cell toxicity test (e.g. MTT) would be more relevant to study toxicity.

Histogram presentation of GFP(+) cells would have been useful to appreciate their distribution within the whole population.

The absence of cytotoxicity mentioned by the authors is not convincing in figure 1-e.  Please comment.

GFP (+) cells show variable fluorescent intensity. Does this mixture allow for an increase in the average number of genomes per cell? Estimating the number of vector genome copies per cell would further characterize the benefit of using the LeB/PB mixture on adenovirus-mediated cell transduction.

Has the effect of the LeB/PB mixture alone on surface marker expression or cytokine secretion been evaluated?

The HAdV-5-GFP vector induces the maturation of iBMDCs unlike the HAdV-5-Luc1 vector. Could a difference in the ratio of infectious particles to viral particles of the two vector productions be a source of this difference?

The authors discuss the immunogenicity of GFP, please comment on this paper: Identification of the murine firefly luciferase-specific CD8 T-cell epitopes (doi.org/10.1038/gt.2008.177).

Major comments:

LINE 66 and 67: the authors associate the low transduction efficiency of murine BMDCs with the lack of natural tropism of human adenovirus type 5 for murine cells.The argument needs to be rephrased, HAdV-5 has no greater affinity for human dendritic cells. Moreover, the efficiency of gene transfer in mice using HAdV-5 derived vectors is no longer in question.

Comments:

LINES 62 - 66: The concept of high titer is not obvious to non-specialists. I would recommend to specify the multiplicity of infection (MOI) considered and to illustrate MOI frequently used to transduce primary APCs.

LINES 71 and 72: can the authors detail the synergistic action of poloxamer 38 and polybrene and their "receptor independent" property

LINE 142: specify the method for determining the VP/ml concentration from the absorbance reading at 260 nm. What are the characteristics (TCID50/ml and PV/ml) of the 2 viral productions used?

LINE 148 and 169: what is the nature of the buffer used?

LINE 282 : add CD83

LINE 283. How is the "median fluorescent intensity" calculated from the "mean +/- SEM"?

LINE 340: is it necessary to mention "median fluorescent intensity"?

Detail the MOCK conditions in the caption of figures 5, 6, 8, S2 and S4

Minor comments:

Prefer the ICTV naming convention to define human adenovirus type 5: HAdV-5

Check carefully the spelling and typography

LINE 37: Delete "human", "human adenovirus" is currently too restrictive

LINE 39: I suggest changing "in gene therapy clinical trials" to "gene and vaccine therapy clinical trials" to be more consistent with the infectious examples chosen

LINE 42: I suggest changing "trigger both a robust cytotoxic T-cell response as well as a humoral response" to "trigger both a humoral response as well as a robust cytotoxic T-cell response

LINE 89: change "infected" to "transduced

LINE 139: delete "equilibrium

LINE 173. Prefer "labelled" to "stained

Reviewer 2 Report

The manuscript entitled ‘Breaking entry- and species barriers: LentiBOOSTR plus Polybrene enables high transduction efficacy of dendritic cells and monocytes by adenovirus 5’ presented by Knippertz et al and authored primarily by Strack is reviewed.

The paper describes enhanced transduction of immature and mature murine and human dendritic cells by adenovirus type 5 complexed with LentiBOOSTR plus Polybrene.

The work is reasonably comprehensive in its design but missed out certain essential features that makes the data point to potential vector issues and experimental design.

A few minor points:

(a). Ad5 with and without transduction enhancing agents such as polybrene, DEAE dextran and/or DOGS has been previously presented for murine cells eg.

  1. Gregory, L.G., Harbottle, R.P., Lawrence, L., Knapton, H.J., Themis, M., Coutelle, C. Enhancement of adenovirus mediated gene transfer to the airways by DEAE dextran and sodium caprate in vivo. 2003 Mol Ther 7(1): 19-26
  2. Gregory LG, Harbottle RP, Lawrence L, Knapton HJ, Themis M, Coutelle C. Enhancement of adenovirus-mediated gene transfer to the airways by DEAE dextran and sodium caprate in vivo. Mol Ther. 2003 Jan;7(1):19-26.
  3. McKay, T. R., MacVinish, L. J., Carpenter, B., Themis. M., Jezzard, S., Goldin, R., Pavirani, A., Hickman, M. E., Cuthbert, A. W., Coutelle, C. Selective in vivo transfection of murine biliary epithelia using polycation-enhanced adenovirus. Gene Ther. 2000 Apr;7(8):644-52.

And therefore it is known that murine cells are easily infected with this vector – line 67. This somewhat alters the title of the paper, which should be more specific please.

(b). The fact that LentiBOOSTR plus Polybrene has been used in the clinic as mentioned in the introduction is not strictly a talking point because this paper uses an increased amount of polybrene, which is known to be toxic to cells and not acceptable in a clinical setting  - line 70.

Major points:

(c). The manuscript mentions LentiBOOSTR plus Polybrene is the enhancer provided by Sirion Biotech, but what amount of polybrene is used in complex before extra polybrene is added for the work here. Can the authors please provide information on the controls. Is LentiBOOSTR (provided by the manufacturer) used as a control with no polybrene at all or with polybrene?

(d). What are the titres of the vectors produced? It appears there are two sources of vector here. One generated in the laboratory of the author (GFP construct) and the other provided by David Curiel (Luc vector). Were both vectors grown in the author’s lab and titred side by side? Were both batches generated by CsCl and dialysed or was the Luc vector produced at greater purity using affinity chromatography. If this is the case, then there may well be differences in titre and/or purity that would explain why the GFP vector gives different results than the Luc vector. Obvious GFP should not be expected to show differences in outcome, so this seems strange. I suspect the GFP Ad5 may not be very pure and causing an antigen response in the paper. Comparison of vector purity and titre is therefore high important here.

(e). Why did the experimental controls not use the following:

LentiBOOSTR/Polybrene (manufacturer’s) as the transduction agent alone plus Ad5

LentiBOOSTR/Polybrene (manufacturer’s) with extra polybrene alone plus Ad5

Ad5 alone with/without different amounts of polybrene (important)? Polybrene with Ad5 alone has been reported previously to enhance transduction of murine cells and I suspect it may have done so in this work rather than the LentiBOOSTR/Polybrene (manufacturer’s) plus additional polybrene conjugate. Hence, this control prevents the levels of transduction being assigned to the LentiBOOSTR/Polybrene (manufacturer’s) plus polybrene and Ad5.

(f). For cell survival, it would be important to provide a more robust survival/death assay. This would be an MTT assay and is needed for the effects of the transduction agents to be clearly assessed.

(g). After infection what was the doubling times over a 7-10 day period. Ie. what are the long term effects of infection?    

Reviewer 3 Report

Adenovirus vectors are commonly used in gene therapy and vaccine studies. Although adenovirus infection and host responses are rather well understood in epithelial-like cells, interactions of the virus with innate immune cells are less well characterized, mainly due to difficulties of infecting these cells in vitro. Strack et al. report that use of LentiBOOST/Polybrene enhances adenovector-mediated transduction of both murine and human primary immature (iDC) and mature (mDC) dendritic cells, as well as human monocytes. Functional analyses of the transduced cells indicated that although LentiBOOST/Polybrene has some effects on human immature dendritic cells, e.g. maturation markers CD83 and CD86 were upregulated in iDCs, and IL-8 production was up as well, in general, no major impairment in dendritic cell or monocyte function was observed with the use of LentiBOOST/Polybrene. Human mDCs retained T cell stimulatory capacity after LentiBOOST/Polybrene + adeno-vector transduction and differentiation of transduced monocytes into functional iDCs and mDCs was intact. Overall, the results described in the manuscript are of interest for researchers working with adenovirus vectors, as well as for basic research into virus interactions with monocytes/dendritic cells.  Some typos in the manuscript has to be corrected and the Authors might consider the points below for minor changes in the manuscript.

Minor points:

1) on page 2 the paragraph between lines 68-81 is devoid of any references although several statements are made in this paragraph

2) it is unclear what the MOI of virus transductions refers to: physical particles/cell or infectious particles/cell. The former would be more informative because infectious particle MOI refers only to the cell line used for titration

3) page 4, line 147: “Virus suspension was prepared using 0.5mg/ml LentiBOOST + 4 ug/ml Polybrene…” What was the number of physical virus particles in the virus suspension, i.e. is the ratio of LentiBOOST/Polybrene to virus particles important for transduction efficiency?

4) Fig.1 is mislabeled as Fig.2

5) Page 6, line 243: “LeB/PB did not affect the morphology of either iBMDCs or mBMDCs (Figure 1e,f)”. This is impossible to see in the images provided, although the images clearly show the increase in the transduction efficiency with LentiBOOST/Polybrene. The Authors could consider providing higher magnification images in the supplementary section. This applies to the Fig. 4 e, f as well.

6) Figure legend to supplementary figure S3: buffer only should be (-LeB/PB), not ( LEB/PB)

7) In general, when one has results from three technical/biological replicates, is bar plot really the best way to show the results? Showing the results as individual dots would be an alternative. For example, presentation of results as a bar plot  in Fig. S4a/IL-8 makes it a bit difficult to believe that the differences in the  –LeB/PB and +LeB/PB Ad5-GFP MOI 100 results are not statistically significant

8) Fig. 7 a and d: the figure legend states that the middle panels show median fluorescence intensity of GFP but the y-axis is annotated as mean GFP. The same in Fig.4.

9) Supplementary figure S5: the figure legend states that the “Data are mean +/- SEM”. Where is the SEM? I assume that the black horizontal bar is the mean. It would be better to present the bar with a different color.

10) The last sentence in the Discussion section “Therefore, using LentiBOOST/Polybrene offers new means for promoting HAdV- or LV mediated viral gene transfer in vitro as well as in vivo for future efficient and safe clinical applications”. This is a bit overstatement at the moment, since only limited (no omics) host response analyses were carried out in the present study, and, unless extensive clinical studies with LentiBOOST/Polybrene have been conducted with lentivectors (the reference list did not list such studies), it is perhaps best at this moment to state that LentiBOOST/Polybrene shows potential in promoting HAdV- mediated gene transfer….